# Natural Small-Molecule Bergapten Ameliorates Amyloid-β Pathology and Neuroinflammation in Alzheimer’s Disease

**DOI:** 10.3390/nu17203218

**Published:** 2025-10-14

**Authors:** Jingyan Zhang, Jing Zhang

**Affiliations:** 1The School of Basic Medical Sciences, Fujian Medical University, Fuzhou 350004, China; jingyanzhang32@gmail.com; 2The Graduate School, Fujian Medical University, Fuzhou 350004, China; 3Fujian Key Laboratory of Molecular Neurology, Institute of Neuroscience, Fujian Medical University, Fuzhou 350004, China

**Keywords:** bergapten, 5×FAD mice, neuroinflammation

## Abstract

Background: The pathogenesis of Alzheimer’s disease (AD) is complex, and effective treatments remain elusive. Growing evidence suggests that dietary factors may play a significant role in preventing or alleviating AD. Bergapten (BG), a natural compound with anti-inflammatory properties, has been studied; however, its specific role in neuroinflammation and AD pathogenesis remains unclear. Methods: Through public databases and bioinformatics tools, the possible molecular mechanisms of BG’s effects on AD were analyzed. Six-month-old 5×FAD mice underwent intragastric administration of BG for 30 consecutive days. Learning and memory abilities were assessed using the novel object recognition (NOR) test and the Morris water maze (MWM) test. Immunofluorescence staining, Western blot and q-PCR was conducted to assess the underlying mechanisms. In vitro experiments used Aβ-stimulated BV2 microglial cells for BG intervention. Results: Bioinformatics analysis revealed the MAPK signaling pathway as the top-ranked pathway. Molecular docking studies further demonstrated strong binding interactions between BG and key proteins within the MAPK pathway. In behavioral studies, NOR test and MWM test demonstrated that BG treatment improved learning and memory abilities in 5×FAD mice. Additionally, BG treatment significantly reduced Aβ deposition, pro-inflammatory cytokine levels, and inhibited excessive microglial activation in these mice. Consistent with in vivo findings, BG effectively decreased pro-inflammatory cytokines in Aβ-stimulated BV2 microglial cells. Mechanistic studies revealed that BG attenuates neuroinflammatory responses by inhibiting the MAPK signaling pathway both in vivo and in vitro. Conclusions: Our findings suggest that BG mitigates AD pathological features by suppressing MAPK-mediated neuroinflammation and represents a promising natural small molecule for the prevention and treatment of AD.

## 1. Introduction

Alzheimer’s disease (AD) is the leading cause of late-onset dementia globally and represents a major public health challenge. Its prevalence is projected to rise significantly with the aging population [1,2,3,4]. The classical pathological hallmarks of AD are amyloid-β (Aβ) plaques and hyperphosphorylated tau accumulation [5,6]. Neuroinflammation, driven by microglia and astrocytes, is also a critical pathological process in AD [6,7,8]. In recent years, several therapeutic drugs targeting Aβ, such as lecanemab and aducanumab, have been developed [4,9]. However, these agents demonstrate limited clinical efficacy in AD patients, largely because they primarily target Aβ removal without effectively mitigating neuroinflammation in the brain [4,10,11,12,13]. This underscores the critical need for more effective therapeutic strategies that specifically target neuroinflammation in AD treatment.

Bergapten (BG), a natural small molecule, is widely distributed in various medicinal plants and common dietary sources [14]. These include fruits such as bergamot, grapefruit, and lime, as well as vegetables like parsley, celery, and carrots [14]. BG exhibits a broad spectrum of pharmacological activities, encompassing anti-depressive [15,16,17,18], anti-inflammatory [19,20,21,22,23,24,25,26,27,28,29], anti-oxidative [30,31,32,33], and anti-tumor effects [34,35,36,37]. Furthermore, pharmacokinetic studies indicate that BG possesses relatively high absolute bioavailability and demonstrates the ability to cross the blood–brain barrier [14,15]. These properties underscore its strong therapeutic potential for neurodegenerative diseases.

Recent studies have investigated the potential neuroprotective effects of BG. BG ameliorates scopolamine-induced memory impairment in mice through cholinergic and antioxidative mechanisms [38]. Administration of BG significantly prolonged the anti-depressant effects induced by nicotine [39]. Furthermore, BG alleviated Aβ-induced depressive behavior in mice by suppressing nuclear factor-κB (NF-κB) signaling and the mitogen-activated protein kinase (MAPK) pathway [16]. Additionally, BG mitigated streptozotocin-induced sporadic Alzheimer’s disease in mice via modulation of neuroinflammation [40]. However, it remains unknown whether long-term BG treatment can attenuate neuroinflammation and exert neuroprotective effects on AD.

Thus, this study employs an integrated approach combining bioinformatics analysis with experimental validation. We first identified the principal pathways through which BG modulates AD pathogenesis, and then investigated the protective effects of long-term BG treatment on cognitive function, Aβ pathology, and neuroinflammatory responses in 5×FAD mice and Aβ-stimulated BV2 cells, while elucidating its underlying anti-inflammatory mechanisms.

## 2. Materials and Methods

### 2.1. Animals

A total of 5×FAD transgenic mice, on a C57BL/6 genetic background, were obtained from Jackson Laboratory (Stock No. 034848-JAX; Bar Harbor, ME, USA). These mice overexpress human amyloid precursor protein (APP 695) carrying the K670N/M671L (Swedish), I716V (Florida), and V717I (London) mutations, as well as human presenilin 1 (PS1) with the M146L and L286V mutations, under the control of the Thy-1 promoter. These 5×FAD hemizygous transgenic males were bred with wild-type (WT) females of the same genetic background. Offspring were genotyped by polymerase chain reaction (PCR) using DNA extracted from tail tissues. The APP-specific primer sequences used were as follows: Forward: 5′-AGA GTA CCA ACT TGC ATG ACT ACG-3′; Reverse: 5′-ATG CTG GAT AAC TGC CTT CTT ATC-3′. Mice were housed at 3–5 animals per cage under a 12 h light/dark cycle at 22 °C with ad libitum access to standard rodent chow and autoclaved water. All animal procedures were approved by the Institutional Animal Care and Use Committee (IACUC) of Fujian Medical University (FJMU-2021-0272 (6 August 2021)) and strictly adhered to the National Institutes of Health (NIH) Guide for the Care and Use of Laboratory Animals.

### 2.2. Drugs

Bergapten (BG; CAS: 484-20-8, purity ≥ 98%) was purchased from Sichuan Vicky Biotechnology (Sichuan, China). BG was suspended in 0.5% carboxymethycellulose sodium (CMC-Na) for in vivo experiment [19,41].

### 2.3. Experimental Design of BG Administration

Six-month-old WT (*n* = 19) and 5×FAD (*n* = 18) female mice were randomly assigned to experimental groups, and blinding was applied during experimentation and data analysis. The BG-treated group (WT + BG, *n* = 9) and the BG-treated AD group (5×FAD + BG, *n* = 9) received daily intragastrical gavage administration of BG (30 mg/kg) for 30 consecutive days, while those in the normal control (WT + Vehicle, *n* = 10) and the AD control (5×FAD + vehicle, *n* = 9) groups were administered an equivalent volume of 0.5% CMC-Na solution daily for the same duration. The body weights of all animals were monitored weekly throughout the study. To account for possible circadian variations in drug response, BG was administered at a fixed time each day (15:00) to ensure rigorous temporal consistency across the experimental period.

### 2.4. Behavior Tests

Prior to experimentation, animals were housed for one week to acclimatize. In accordance with the 3Rs principle (Replacement, Reduction, Refinement), experimental procedures were designed to minimize the number of animals used and to alleviate any potential suffering.

#### 2.4.1. Novel Object Recognition (NOR) Test

The NOR test, widely used to assess object recognition memory [42], exploits rodents’ innate preference for novel objects. This test was conducted in a white acrylic open-field arena (40 × 40 × 40 cm). The procedure comprised four sequential sessions: habituation, training, and two testing sessions. The NOR task evaluates both short-term and long-term memory by manipulating the retention interval—the time animals retain memory of training objects. On day 1, each mouse underwent a 5 min habituation session in the empty arena. On day 2, during the training session, mice freely explored two identical objects (placed near adjacent corners) for 5 min after being released facing away from them at the center of the opposite wall. Following a 2 h retention interval, Testing Session 1 introduced a novel object (N1); mice explored both familiar and novel objects for 5 min (novel/familiar object locations counterbalanced). Testing Session 2 occurred after a 24 h retention interval using a second novel object (N2). Exploration time for novel versus familiar objects was recorded. Novel object preference, reflecting discrimination ability, served as an index for short-term (2 h interval) or long-term (24 h interval) recognition memory. The discrimination index was calculated as: (Time exploring novel object − Time spent exploring familiar object)/Total object exploration time. This index provided a normalized measure of novelty preference.

#### 2.4.2. Morris Water Maze (MWM) Test

This behavioral test primarily assesses spatial learning and memory in mice, conducted according to established protocols [43]. Briefly, testing occurred in a dark circular pool (diameter: 1.2 m, height: 0.5 m) filled with opaque, whitened water (22 ± 1 °C) to a depth of 35 cm, with distinctly shaped visual cues positioned around the room walls. A fixed escape platform (10 cm diameter), submerged 1.5 cm below the water surface, was placed in the center of one quadrant. During the 5-day training phase, each mouse underwent four trials daily; each trial commenced from one of four starting positions according to a semi-random sequence [43], with the mouse facing the pool wall upon release and allowed 60 s to locate the hidden platform. Mice failing to find the platform within this time were gently guided to it and remained for 20 s, with escape latency calculated as the average time across four trials per session. Twenty-four hours post-training, a probe test assessed memory retention: the platform was removed, and each mouse swam freely for 60 s while the time spent in the target quadrant and the number of crossings over the former platform location were recorded. Swimming paths were tracked and analyzed using EthoVision XT video tracking software (EthoVision XT 16, Noldus Information Technology, Wageningen, The Netherlands).

### 2.5. Preparation of Oligomeric Aβ1–42 and BG

Aβ1-42 peptide powder (AnaSpec, Cat# AS-20276, Fremont, CA, USA) was initially dissolved in ice-cold hexafluoroisopropanol (HFIP) [44]. The solution was evaporated to dryness to remove HFIP. The resulting peptide film was then reconstituted in dimethyl sulfoxide (DMSO) to a stock concentration of 5 mM, followed by dilution in sterile phosphate-buffered saline (PBS) to 100 μM. This solution was incubated at 4 °C for 24 h to facilitate oligomerization. After centrifugation at 14,000× *g* for 10 min (4 °C) to remove insoluble aggregates, the supernatant containing soluble oligomers was collected and further incubated at 37 °C for 16 h prior to experimental use.

BG was dissolved in DMSO and serially diluted in DMEM culture medium to achieve target working concentrations.

### 2.6. Cell Culture and Treatment

#### 2.6.1. BV2 Cells Culture and Treatment

The murine microglial BV2 cell line was purchased from Pricella (Cat#CL-0493, Wuhan, China) and cultured in BV2 Specialty Medium (Pricella, Cat# CM-0493, Wuhan, China), maintained at 37 °C in a 5% CO_2_ incubator. Cells were pretreated with or without BG (30, 50, and 100 μM) for 6 h, followed by Aβ (100 nM) stimulation for 24 h [45]. After treatment, cells were harvested for subsequent analysis.

#### 2.6.2. Cell-Counting Kit-8 (CCK-8) Assay

BV2 cells were seeded in 96-well plates and cultured for 24 h. To assess cell viability after treatment with BG at varying concentrations (0–100 μM), a CCK-8 assay was performed at 48 h according to the manufacturer’s protocol (GLPBIO, Cat# GK10001, Montclair, CA, USA).

### 2.7. Immunofluorescence

Mice were transcardially perfused under anesthesia via the ascending aorta with 0.1 M PBS followed by 4% paraformaldehyde (PFA). Brains were isolated, post-fixed in 4% PFA overnight, and then cryoprotected in 30% sucrose (*w*/*v*) phosphate buffer. Coronal sections (40 μm) were cut on a cryostat and stored in cryoprotectant. Free-floating sections underwent permeabilization in 0.5% Triton X-100/TBS (1 h), followed by blocking in 10% goat serum/0.5% Triton X-100/TBS (1.5 h, RT). Sections were incubated overnight at 4 °C with primary antibodies in blocking buffer: rabbit anti-GFAP (Abcam, Cat# ab4674, Cambridge, UK), Iba1 (Wako, Cat# 019-19741, Osaka, Japan), or anti-Aβ (6E10, BioLegend, Cat# 803001, San Diego, CA, USA). After TBS washes, sections were incubated with corresponding secondary antibodies (1:2000 dilution, 2 h, RT). Images were acquired using a Zeiss confocal microscope (LSM880) and quantified with Fiji software (Java 1.8.0_322).

### 2.8. RNA Extraction and Quantitative Real-Time PCR (qRT-PCR)

Adult mice were euthanized by decapitation and the hippocampus was immediately dissected on ice. All samples were snap-frozen in liquid nitrogen and stored at −80 °C until RNA extraction using the RaPure Total RNA Kit (Magen, Cat#R401103, Guangzhou, China). cDNA was synthesized with the HiScript III All-in-one RT SuperMix Perfect for qPCR Kit (Vazyme, R333-01, Nanjing, China), followed by real-time PCR quantification on a QuantStudio 5 system (Thermo Fisher, Waltham, MA, USA). Primer sequences for pro-inflammatory cytokines (Il-1β, Il-6, Tnf-α) were Il-1β, forward-5′-GCAACTGTTCCTGAACTCAACT-3′, reversed-5-ATCTTTTGGGGTCCGTCAACT-3′; Il-6, forward-5-TAGTCCTTCCTACCCCAATTTCC-3′, reverse--TTGGTCCTTAGCCACTCCTTC-3′; Tnf-α, forward-5′-CCCTCACACTCAGATCATCTTCT-3′, reverse-5′-GCTACGACGTGGGCTACAG-3′; β-actin, forward-5′-GATCATTGCTCCTCCTGAGC-3′, reverse-5′-ACTCCTGCTTGCTGATCCAC-3′. Relative gene expression fold-changes were calculated using the 2−ΔΔCT method normalized to β-actin.

### 2.9. Western Blot 

Frozen hippocampal tissues were homogenized in RIPA lysis buffer (CST, #8553) supplemented with a protease inhibitor cocktail (MCE, HYK0010), phosphatase inhibitors (MCE, HYK0022), and PMSF (CST, #8553). Homogenates were sonicated on ice (100 W, 1 s on/2 s off, 1 min) and subsequently lysed on ice for 30 min, followed by centrifugation at 12,000× *g* for 20 min at 4 °C. The supernatants were collected after centrifugation. Protein concentration was determined using a BCA assay kit (Beyotime, cat# P0009, Shanghai, China) and adjusted to equal concentrations. Following heat denaturation, equal amounts of protein were separated by SDS-PAGE and transferred onto polyvinylidene fluoride (PVDF) membranes (Millipore, cat#IPVH0000, Burlington, MA, USA). Membranes were blocked with 5% bovine serum albumin (BSA) in Tris-buffered saline containing 0.1% Tween-20 (TBST) for 1 h at room temperature and then incubated with the designated primary antibodies overnight at 4 °C. After washing three times with TBST, membranes were incubated with horseradish peroxidase (HRP)-conjugated secondary antibodies. Immunoreactive proteins were detected using an enhanced chemiluminescent (ECL) substrate (MCE, HY-K1005) and visualized with a Chemiluminescence Imager (eBLOT, Tanon, Xiamen, China). The primary antibodies used included the following: Iba1 (1:2000, Novus, NB100-1028), GFAP (1:2000, SY5Y, 173011), rabbit Phospho-p44/42 MAPK (Erk1/2) (Thr202/Tyr204) (1:1000, CST, #4370), rabbit p44/42 MAPK (Erk1/2) (1:1000, CST, #4695), anti-APP (1:5000, Millipore, 171610), β-actin (1:5000, Abcam, ab8226), and GAPDH (1:5000, Proteintech, 60004-1-AP). Band intensities were quantified using ImageJ software (Java 1.8.0_322), and the number below each band represents the average ratio of the target protein or modification relative to the loading control.

### 2.10. Bioinformatics Analysis

#### 2.10.1. Target Screening of AD and BG

The GeneCards disease databases were searched using “Alzheimer’s disease” as keywords to retrieve related targets of AD. Subsequently, the InChIKey information (BGEBZHIAGXMEMV-UHFFFAOYSA-N) for BG was obtained from the Traditional Chinese Medicine Systems Pharmacology (TCMSP) database; this InChIKey was then used to acquire the corresponding SMILES information (COC1=C2C=CC(=O)OC2=CC3=C1C=CO3) from the PubChem database. The SMILES data was entered into the Swiss Target Prediction database [46] for target prediction. Based on the screening results, targets with anAim-listed probability >0 were selected, yielding the predicted targets of BG. Additionally, the targets of BG listed in the Drug-Gene Interaction Database (DGIdb) were retrieved, along with their known drug targets.

#### 2.10.2. Gene Ontology (GO) and Kyoto Encyclopedia of Genes and Genomes (KEGG) Pathway Enrichment Analysis

The predicted targets of BG and AD were intersected using the free online platform OmicShare (http://www.omicshare.com/tools/ (accessed on 1 June 2025)) to generate a Venn diagram. Subsequently, core targets derived from this intersection underwent GO and KEGG pathway enrichment analyses, also performed using OmicShare tools (a Zero-Code Interactive Online Platform for Biological Data Analysis and Visualization, iMeta e228). GO analysis predicted gene function across three categories: Molecular Function (MF), Biological Process (BP), and Cellular Component (CC). A *p*-value < 0.05 was considered statistically significant, with specific statistical details provided in the figure legends. The top 10 significantly enriched GO terms and the top 20 KEGG pathways were selected for visualization via bubble plots.

#### 2.10.3. Protein–Protein Interaction (PPI) Network Analysis

The PPI network of the obtained intersection targets was constructed using the STRING 12.0 database (https://cn.string-db.org/), with the minimum interaction score set to 0.4. This network was then imported into Cytoscape software (version 3.10.3), and its topological parameters (including degree and betweenness centrality) were analyzed using the CytoNCA (2.0) plugin. These analyses aimed to identify important nodes (such as key proteins within the MAPK signaling pathway) and explore potential functional modules within the network through maximum clique-based module mining. For visualization, node importance was represented by color intensity based on Degree value, with warmer (darker red) hues indicating higher importance within the PPI network.

#### 2.10.4. Molecular Docking Simulation

To analyze the binding affinities and interaction modes between BG and its targets, molecular docking was performed using AutoDock Vina 1.2.2, an in silico protein–ligand docking software. The 3D coordinates of the target proteins NFκB1 (PDB code: 2O61), PTK2 (PDB code: 4NY0), MET (PDB code: 2UZY), PDGFRB (PDB code: 3MJG), and MAP3K8 (PDB code: 4Y83) were retrieved from the RCSB Protein Data Bank (PDB). The molecular structure of BG (PubChem CID: 2355) was obtained from the PubChem database. Docking simulations were conducted using AutoDock Vina 1.2.2, and the resulting complexes were visualized with PyMOL(2.6.2) software.

### 2.11. Statistical Analysis

Data are presented as mean ± SEM. All statistical analyses were performed using GraphPad Prism 9 (GraphPad Software, www.graphpad.com, San Diego, CA, USA). Appropriate statistical tests were selected based on experimental design. Comparisons between two groups were analyzed using Student’s *t*-test. For comparisons among multiple groups with homogeneous variance, one-way ANOVA followed by Tukey’s post hoc test was applied to identify intergroup differences. Escape latencies in the MWM and body weight were analyzed using two-way ANOVA. When variance heterogeneity was detected, Welch ANOVA tests were used, followed by Dunnett’s test for multiple comparisons. If the data are not normally distributed, the Kruskal–Wallis test should be employed for analysis. Outliers, defined as data points lying beyond the mean ± 2 standard deviations (SD), were excluded from the analysis. A *p*-value of less than 0.05 was considered statistically significant.

## 3. Results

### 3.1. Analysis of Publicly Available Databases Revealed That BG Modulates the MAPK Signaling Pathway in AD

Following the experimental design outlined in Figure 1A, we identified overlapping targets between AD and BG using public databases. Specifically, we searched GeneCards using “Alzheimer’s disease” as the keyword, retrieving 15,203 related genes. For BG (chemical structure shown in Figure 1B), we obtained 51 predicted targets: 45 with Probability > 0 from SwissTargetPrediction and 6 from the Drug-Gene Interaction Database (DGIdb). Intersection analysis yielded 50 overlapping targets (Figure 1C), supporting a potential association between BG and AD.

Subsequent GO and KEGG enrichment analyses were performed on these 50 targets. The top 10 most significantly enriched terms (based on smallest *p*-value) for BP, CC, and MF were visualized using enrichment circle diagrams (Figure 1D and Appendix A). Circular visualization indicated that BG may alleviate AD primarily through modulating metabolic processes. KEGG pathway analysis further revealed that numerous overlapping targets are associated with neuroinflammation pathways, including the MAPK signaling pathway, arachidonic acid metabolism, and the serotonergic synapse (Figure 1E). Notably, the MAPK signaling pathway exhibited the most significant alterations (highlighted by the red box in Figure 1E).

Collectively, these findings demonstrate that BG modulates the MAPK signaling pathway in AD.

### 3.2. PPI Network Analysis and Molecular Docking Suggested BG’s Modulation of MAPK Signaling Pathway Proteins

To further elucidate the mechanism of BG in alleviating AD, we constructed a protein–protein interaction (PPI) network. To enhance clarity regarding the regulatory roles of the MAPK signaling pathway and other key pathways within the PPI network, we segregated them into distinct subnetworks (left and right) using Cytoscape (v3.10.3) (Figure 2A). Analysis of the left subnetwork (representing the MAPK pathway) identified four core targets (NF-κB1, PTK2, MET, PDGFRB) with high degree values, indicating their central regulatory roles. Additional core targets with high degree values in other subnetworks may also play significant regulatory roles.

We next investigated whether BG could directly bind to the key MAPK-related proteins NF-κB1, PTK2, MET, and PDGFRB using molecular docking simulations. The results demonstrated specific binding interactions: NF-κB1: BG formed a hydrogen bond with Arg2031; PTK2: BG formed a hydrogen bond with Ala81. MET: BG formed a hydrogen bond with Ser531; PDGFRB: BG formed hydrogen bonds with Ser187, Cys16 and Glu15. Given the critical importance of MAP3K8 within the MAPK signaling pathway, we also docked BG to this protein. BG formed hydrogen bonds with Gly210 and Arg146 of MAP3K8 (Figure 2B).

Collectively, these molecular docking studies suggested BG’s ability to directly interact with and modulate key molecular targets within the MAPK signaling pathway, including NF-κB1, PTK2, MET, PDGFRB and MAP3K8, providing insights into the binding modes and specific interactions.

### 3.3. BG Treatment Ameliorates Cognitive Deficits in 5×FAD Mice

To investigate whether BG treatment ameliorates cognitive deficits in 5×FAD mice, we administered BG once daily for 30 days. Learning and memory were then assessed at 6 months of age using the NOR and MWM tests (Figure 3A).

The NOR test, performed as described [42] (Figure 3B), evaluates recognition memory. During the short-term memory test (2 h retention interval), WT mice spent significantly more time exploring the novel object than the familiar object (*p* < 0.0001). In contrast, untreated 5×FAD mice showed no preference (*p* = 0.272152). Importantly, BG-treated 5×FAD mice exhibited a significant preference for the novel object (*p* < 0.01, Figure 3C, left panel). The discrimination index (DI), a quantitative measure of recognition memory, was significantly lower in 5×FAD mice compared to WT controls (*p* < 0.0001). BG treatment significantly reversed this deficit (*p* < 0.01, Figure 3C, right panel). Long-term recognition memory was assessed after a 24 h retention interval. Untreated 5×FAD mice again failed to distinguish the novel object (*p* = 0.753649), whereas WT mice showed a clear preference (*p* < 0.01). Notably, BG-treated 5×FAD mice spent significantly more time exploring the novel object (*p* < 0.05, Figure 3D). Consistent with the findings on exploration time, BG treatment restored the significantly reduced DI in untreated 5×FAD mice (5×FAD + vehicle vs. WT + vehicle, *p* < 0.001; 5×FAD + BG vs. 5×FAD + vehicle, *p* < 0.05). Collectively, these findings demonstrate that BG treatment rescues both short-term and long-term object recognition memory deficits in 5×FAD mice.

In the MWM test, 5×FAD mice not only exhibited significantly longer escape latencies during training days compared to WT controls (*p* < 0.0001) but also benefited from BG treatment, which substantially shortened their latencies (*p* < 0.01; Figure 3E). Moreover, during the probe trial, BG-treated 5×FAD mice made significantly more target entries (*p* < 0.05; Figure 3F,G), collectively demonstrating that BG treatment enhanced spatial learning and memory retention. Importantly, these cognitive improvements were independent of motor function or metabolic alterations, as evidenced by comparable body weights (*p* = 0.3210) and swimming speeds (*p* = 0.6048) across all experimental groups (the bottom-right panel of Figure 3F,H).

Collectively, these results demonstrate that BG treatment rescues recognition and spatial memory impairments in 5×FAD mice.

### 3.4. BG Attenuates Aβ Burden and Microglial Activation in 5×FAD Mice

To assess the impact of BG on Aβ deposition, we performed immunofluorescence analysis of brain sections from 5×FAD mice using the 6E10 antibody. Significantly fewer Aβ plaques were observed in the hippocampus and cortex of BG-treated 5×FAD mice compared to untreated 5×FAD controls (*p* < 0.05 in DG, *p* < 0.05 in CA1, *p* < 0.01 in cortex, Figure 4A), indicating that BG treatment markedly reduces amyloid burden.

The presence of activated microglia and astrocytes surrounding Aβ plaques is a typical feature in AD brains [47]. Aβ plaque-associated microglia and astrocytes drive neuroinflammation through pro-inflammatory cytokine release in AD [6]. To determine BG’s cell-type-specific effects in 5×FAD mice, we assessed activation markers Iba-1 (microglia) and GFAP (astrocytes) via immunofluorescence and Western Blotting.

5×FAD mice exhibited significant microglial activation in hippocampal and cortical regions, evidenced by increased Iba-1 immunofluorescence intensity and morphological hypertrophy (soma enlargement, process retraction). Critically, BG treatment significantly reduced Iba-1 immunofluorescence (*p* < 0.05 in DG, *p* < 0.01 in CA1, *p* < 0.05 in cortex, Figure 4B) and downregulated hippocampal Iba-1 protein levels (*p* < 0.0001, Figure 4C).

Conversely, 5×FAD mice showed robust astrocyte activation (elevated GFAP intensity and enlarged processes). Notably, BG treatment neither altered GFAP fluorescence intensity (*p* = 0.9809 in DG, *p* = 0.9845 in CA1, *p* = 0.9008 in cortex, Figure 5A) nor modulated hippocampal GFAP protein expression (*p* = 0.1004, Figure 5B), indicating no suppression of astrogliosis.

These data demonstrate that BG specifically attenuates microglial—but not astrocytic—activation, defining its anti-inflammatory mechanism in AD pathology.

### 3.5. BG Reduced Pro-Inflammatory Cytokine Expression in 5×FAD Mice and Aβ-Stimulated BV2 Cells

Given that chronic neuroinflammation—a core pathological feature of AD alongside Aβ accumulation—drives disease progression, we next assessed BG’s modulation of pro-inflammatory cytokine expression. Q-PCR analysis demonstrated significant hippocampal upregulation of *Il-6* (*p* < 0.0001 in hippocampus, *p* < 0.0001 in the cortex), *Tnf-α* (*p* < 0.01 in hippocampus, *p* < 0.0001 in the cortex) and *Il-1β* (*p* < 0.05 in hippocampus, *p* < 0.0001 in cortex) mRNA in 5×FAD mice versus WT controls. Notably, BG treatment significantly attenuated expression of *Il-6* (*p* < 0.0001 in hippocampus, *p* < 0.001 in the cortex) and *Tnf-α* mRNA (*p* < 0.05 in hippocampus, *p* < 0.001 in cortex, Figure 6A).

To investigate BG’s effects on microglia-mediated inflammation, we modeled Aβ activation in vitro. CCK-8 assays confirmed that BG (1–100 μM) exhibited no cytotoxicity in BV2 cells (*p* = 0.5386, Figure 6B). Q-PCR revealed Aβ significantly upregulated pro-inflammatory cytokines (*Il-6* (*p* < 0.0001), *Tnf-α* (*p* < 0.01) and *Il-1β* (*p* < 0.01)) in BV2 microglia. Critically, BG dose-dependently suppressed this response *Il-6* (*p* < 0.0001 at 30 μM, *p* < 0.0001 at 50 μM), *Tnf-α* (*p* < 0.05 at 30 μM, *p* < 0.01 at 50 μM) and *Il-1β* (*p* < 0.01 at 30 μM, *p* < 0.01 at 50 μM), with minimal efficacy at 30 μM (Figure 6C). This optimal concentration was therefore selected for subsequent experiments.

Collectively, these findings establish BG’s anti-inflammatory action in Aβ-stimulated microglia via cytokine downregulation.

### 3.6. BG Attenuates Microglial Activation by Suppressing MAPK Signaling

To elucidate BG’s mechanism in regulating microglial activation, we assessed MAPK pathway proteins via Western blot. BG administration attenuated hippocampal p-ERK1/2 (Thr202/Tyr204)/ERK1/2 elevation in 5×FAD mice (*p* < 0.05, Figure 7A). Consistent with in vivo findings, in Aβ-stimulated BV2 microglia, and BG significantly suppressed p-ERK1/2 (Thr202/Tyr204)/ERK1/2 expression (*p* < 0.05, Figure 7B). Collectively, these results demonstrate BG ameliorates neuroinflammation through MAPK pathway inhibition.

## 4. Discussion

In this study, integrated database analysis revealed a significant association between BG and AD, indicating the neuroprotective role of BG mediated through the MAPK signaling pathway. PPI analysis and molecular docking further validated these findings. Human epidemiological studies indicate that women are at a significantly higher risk of developing Alzheimer’s disease than men [48,49]. Correspondingly, female 5×FAD mice demonstrate heightened amyloid-beta plaque pathology compared to age-matched males [50,51]. In addition, aggressive behavior among adult male mice induces stress, which may alter neuroinflammatory and immune responses—thereby introducing confounding variables in studies investigating neuroinflammation [52]. Therefore, to enhance experimental consistency and minimize potential confounds, we selected female mice for this study. We found that long-term intragastric administration of BG ameliorated cognitive impairment, reduced Aβ deposition, and attenuated neuroinflammation in 5×FAD female mice. Our results also demonstrated that BG suppresses pro-inflammatory cytokine production by inhibiting excessive microglial activation. Subsequent mechanistic investigation revealed that BG attenuates neuroinflammatory responses by inhibiting the MAPK signaling pathway, evidenced by significantly reduced phosphorylation of ERK (a key MAPK pathway component) in both the hippocampus of 5×FAD mice and Aβ-stimulated BV2 microglial cells.

The presence of elevated inflammatory cytokines and chemokines in AD patients, along with identified AD risk genes linked to innate immunity, suggests that neuroinflammation plays a key role in AD pathogenesis [11,53,54]. Current spatial transcriptomics have found that complex inflammatory-like changes are observed around AD amyloid plaques, manifested as complex changes in gene expression profiles [55]. Neuroinflammation is a double-edged sword for the AD brain: it clears deposited Aβ and produces cytotoxic substances that exacerbate Aβ deposition and lead to cognitive decline [6,8]. Many naturally occurring phytochemicals can counteract cognitive decline by reducing the expression of peripheral and neuroinflammatory responses, and the activation of microglia and astrocytes [45,56,57]. Considering the diverse functions of microglia, it would be advantageous to use medications that selectively modulate their activity—reducing harmful aspects like neurotoxicity and proinflammation while sparing protective functions such as the phagocytic removal of toxins [58]. Our results have shown that long-term BG treatment ameliorated Aβ deposition and microglial activation in 5×FAD mice.

The presence of elevated inflammatory cytokines and chemokines in AD patients, along with identified AD risk genes linked to innate immunity, suggests that neuroinflammation plays a key role in AD pathogenesis [11,48,49]. Current spatial transcriptomics have found that complex inflammatory-like changes are observed around AD amyloid plaques, manifested as complex changes in gene expression profiles [50]. BG, a natural pharmaceutical monomer, exhibits broad anti-inflammatory activity across disease models. Intraperitoneal administration (10 mg/kg, once daily ×14 days) attenuated vincristine-induced peripheral neuropathy by suppressing inflammatory cytokines and NF-κB signaling [59]. Intragastric delivery (30 mg/kg) inhibited airway inflammation via NR4A1 targeting. An inhalable bioactive lipid nanomedicine incorporating BG demonstrated efficacy in targeted acute lung injury therapy by reducing pro-inflammatory M1-type macrophage polarization [22]. Separately, intranasal BG administration (3–30 mg/kg) effectively ameliorated PM2.5-induced combined allergic rhinitis and asthma syndrome through the suppression of STAT3 and MAPK activation [60]. Notably, 4-day intraperitoneal BG treatment (10 mg/kg/day) significantly alleviated Aβ-induced depression-like behavior by inhibiting cyclooxygenase-2 activity and NF-κB/MAPK signaling in microglia [16]. This study investigated whether prolonged BG treatment (30 mg/kg, intragastric, once daily ×30 days) inhibits glial activation and alleviates neuroinflammation in vivo—an unexplored therapeutic dimension. Our results demonstrated significantly reduced microglial activation and decreased Il-6, Tnf-α and Il-1β mRNA levels in BG-treated 5×FAD mice. Complementing these findings, in vitro experiments demonstrated that BG significantly suppresses the mRNA expression of Il-6, Tnf-α and Il-1β in Aβ-stimulated BV2 cells. Collectively, these results establish that prolonged BG treatment exerts potent anti-inflammatory effects in AD models.

The MAPK family of serine–threonine protein kinases regulates diverse cellular processes, including proliferation, differentiation, apoptosis, inflammation, and innate immunity [61]. Chronic activation of this pathway has been demonstrated in studies using Aβ-overexpressing transgenic mouse models and postmortem hippocampal tissue from AD patients, where elevated levels of phosphorylated ERK (p-ERK) correlate with disease progression [62,63]. As a key mechanism driving enhanced neuroinflammatory responses in AD, the MAPK signaling pathway has emerged as a promising therapeutic target [8,64]. Several inhibitors targeting the MEK/ERK axis, including both pharmacological agents and natural extracts, have shown potential for alleviating AD-related neuroinflammation [65]. Our database analysis identified the MAPK pathway as central to the protective effects of BG in AD. Protein–protein interaction analysis and molecular docking further revealed BG’s binding affinity with core MAPK signaling components—NF-κB1, PTK2, MET, PDGFRB, and MAP3K8. Notably, MAP3K8 acts as a MAPK kinase kinase (MAP3K) that bridges upstream signals to the MEK/ERK cascade and plays a crucial role in regulating immune and inflammatory responses [63]. To functionally validate these predictions, we performed Western blotting in both in vivo and in vitro models. We observed significant increases in p-ERK levels in the hippocampus of 5×FAD mice and in Aβ-treated BV2 microglia. Conversely, BG treatment potently suppressed p-ERK activation, indicating effective inhibition of this canonical pathway. Together, these findings demonstrate that BG ameliorates neuroinflammation in AD through selective suppression of the MAPK signaling cascade.

This study has several limitations. First, while 5×FAD transgenic mice model amyloid-β pathology, they incompletely recapitulate the full spectrum of Alzheimer’s disease pathophysiology. Thus, future studies should evaluate BG’s anti-inflammatory effects in tauopathy models (e.g., PS19 mice) for comprehensive validation. Second, the phototoxicity of bergapten, a side effect, should be avoided during anti-inflammatory treatment for Alzheimer’s disease [14]. Third, we cannot exclude potential anti-inflammatory effects mediated by additional signaling pathways beyond those identified here. To definitively establish causality, future studies should employ targeted interventions against MAPK signaling. A key experiment would be to co-administer bergapten with a specific MAPK pathway inhibitor in the 5×FAD model. Fourth, future research should evaluate the long-term safety, efficacy, and mechanistic pathways of BG in both sexes and diverse drug administration systems.

## 5. Conclusions

This study provides preclinical evidence that BG may mitigate amyloid-β pathology by reducing neuroinflammation. These effects appear to be mediated, at least in part, by the inhibition of MAPK signaling in the hippocampus. While our results offer preliminary insights into the neuroprotective properties of BG, its clinical relevance remains to be established.

## Figures and Tables

**Figure 1 nutrients-17-03218-f001:**
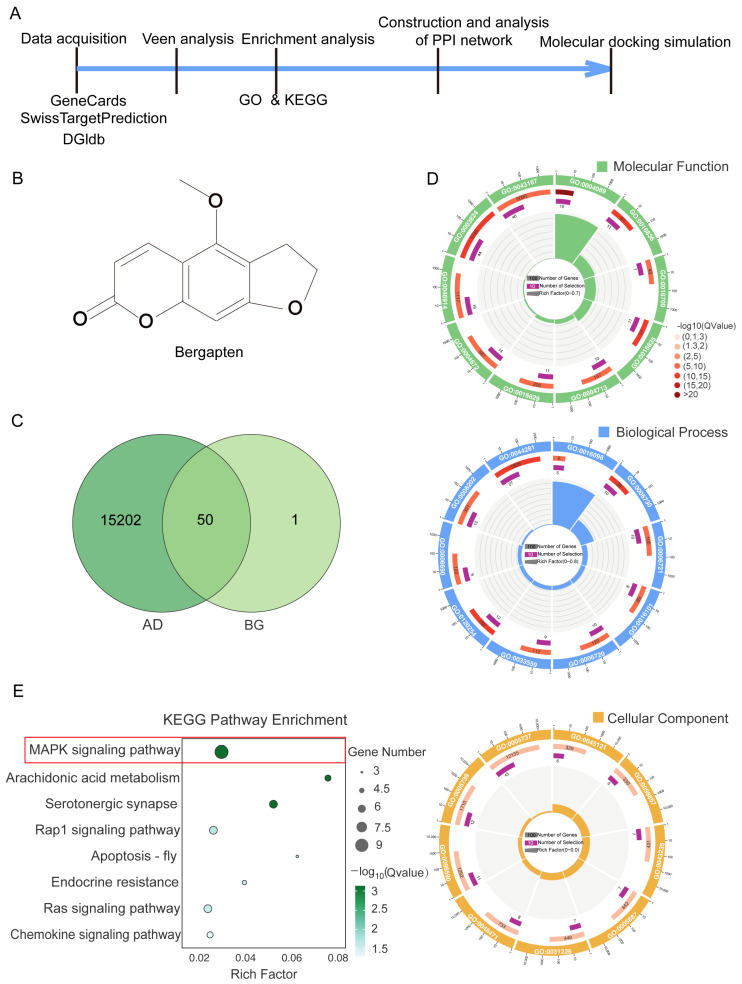
Analyzing public databases demonstrated That BG modulates the MAPK signaling pathway in AD. (**A**) Experimental design of bio-informatics analysis experiment design. The blue arrow represents the timeline. (**B**) The chemical structure of BG. (**C**) The Venn diagram shows the overlap target genes between AD (dark green) and BG (light green). (**D**) The Circle diagram shows top 10 signaling pathways ranked by Q-value after GO-MF (top panel), GO-BP (middle panel), and GO-CC (bottom panel) enrichment analysis (targets significant at the Q-value  ≤  0.05 of 50 overlap targets). (**E**) Bubble plot shows KEGG enrichment analysis of 50 overlap target genes. The red box marks the MAPK signaling pathway.

**Figure 2 nutrients-17-03218-f002:**
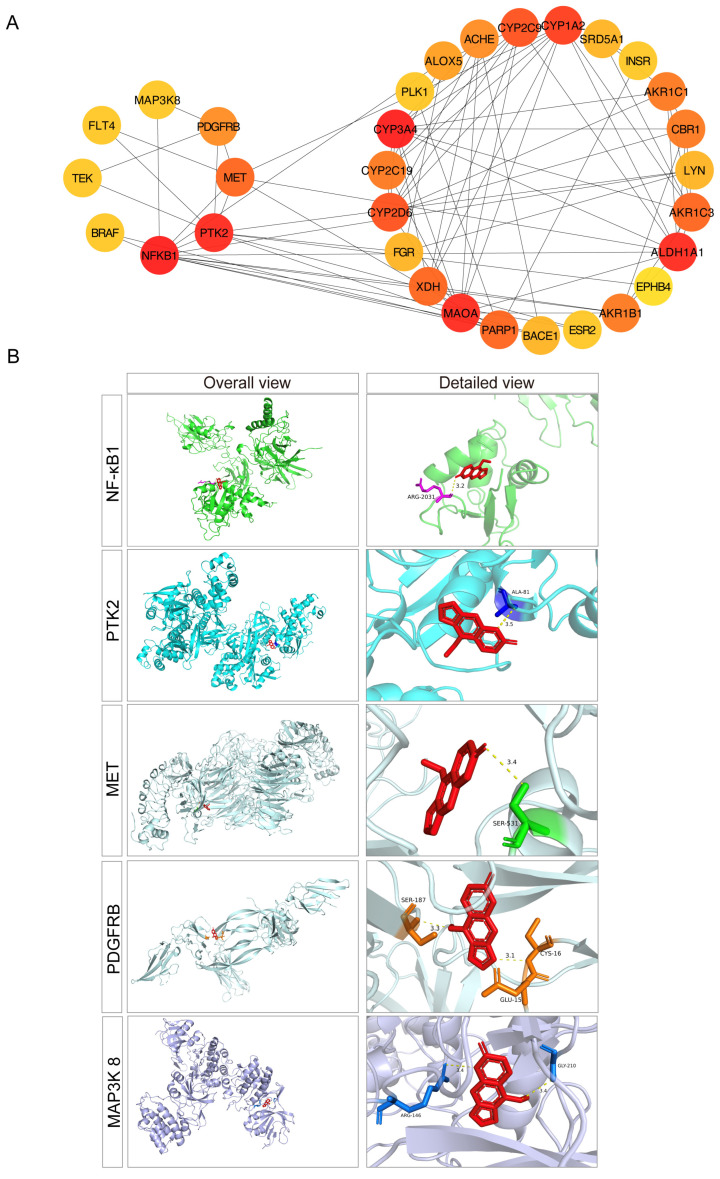
The verification of PPI network and molecular docking simulation. (**A**) Construction of PPI network diagram. Left shows the protein in MAPK signaling pathway. Right shows the protein in other signaling pathway. node importance was represented by color intensity based on Degree value, with warmer (darker red) hues indicating higher importance within the PPI network. (**B**) Schematic of the molecular docking simulation results of BG and NF-κB1, PTK2, MET, PDGFRB and MAP3K8 proteins (from **top** to **bottom**). The yellow dotted line represents the length of the hydrogen bond. The highlighted sticks represent the names of the bound amino acid residues.

**Figure 3 nutrients-17-03218-f003:**
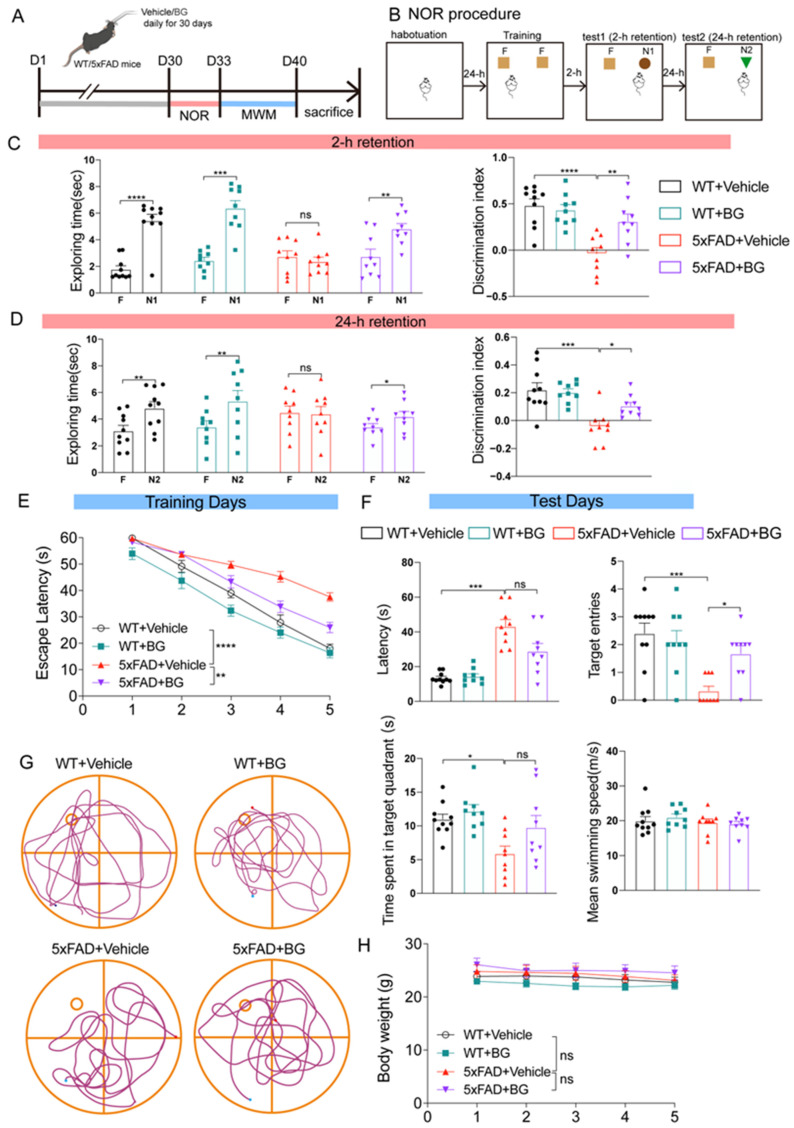
BG attenuated learning and memory deficits in 5×FAD mice (**A**) Schematic diagram of intragastric administration of BG. (**B**) Schematic of the experimental procedure of the NOR test. (**C**) Trial 1 with a 2 h retention interval. **Left**, exploration time between familiar (F) and novel object 1 (N1). **Right**, discrimination index. (**D**) Trial 2 with a 24 h retention interval. **Left**, exploration time between familiar (F) and novel object 2 (N2). **Right**, discrimination index. (**E**) Escape latency to the platform during the training trials in an MWM. (**F**) The latency of first time to enter the target, target entries, time spent in target quadrant, and the mean swimming speed of mice in the probe trial of MWM. (**G**) Representative track images of mice in the probe trial of MWM. (**H**) Body weight analysis on a weekly basis (from the start of drug administration to the end of behavioral testing). *n* = 10, 9, 9, and 9 mice for WT + vehicle, WT + BG, 5×FAD + vehicle and 5×FAD + BG, respectively, in NOR test, MWM and body weight analysis. Data are mean ± SEM. * *p* < 0.05, ** *p* < 0.01, *** *p* < 0.001 and **** *p* < 0.0001; ns, not significant. Two-tailed paired Student’s *t* test (left panel in (**C**) and left panel in (**D**)) or one-way (right panel in (**C**) and right panel in (**D)**, upper right panel, left lower panel, right lower panel in **F**) or two-way (**E**,**H**) ANOVA, followed by Tukey’s multiple comparisons test. Welch’s ANOVA test, followed by Dunnett’s T3 multiple comparisons test (upper left panel in **F**).

**Figure 4 nutrients-17-03218-f004:**
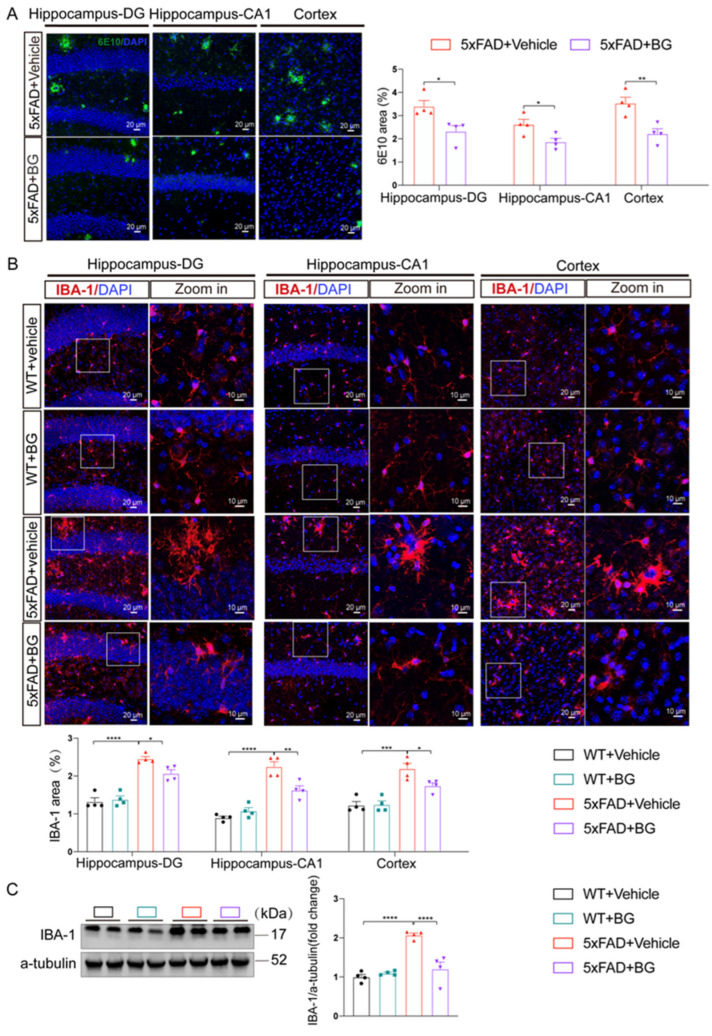
BG reduced Aβ plaque deposition and reduced activated microglia in 5×FAD mice. (**A**) Representative images of 6E10-labeled Aβ plaques and quantification of immunoreactivity area of 6E10 in hippocampal DG region, CA1 region, and cortex of 5×FAD + vehicle and 5×FAD + BG, respectively (*n* = 4 mice per group). The white frame represents the area to be zoomed. (**B**) Representative images of Iba-1-labeled microglia and quantification of immunoreactivity area of Iba-1 in hippocampal DG region, CA1 region, and cortex of WT+vehicle, WT+BG, AD +vehicle and 5×FAD+BG, respectively (*n* = 4 mice per group). (**C**) Representative immunoblots and quantitative analyses of Iba-1 in hippocampus of WT + vehicle, WT + BG, 5×FAD + vehicle and 5×FAD + BG *(n* = 4 per group). Data are mean ± SEM. * *p* < 0.05, ** *p* < 0.01, *** *p* < 0.001 and **** *p* < 0.0001; Two-tailed unpaired Student’s *t* test in (**A**). One-way ANOVA, followed by Tukey’s multiple comparisons test in (**B**,**C**).

**Figure 5 nutrients-17-03218-f005:**
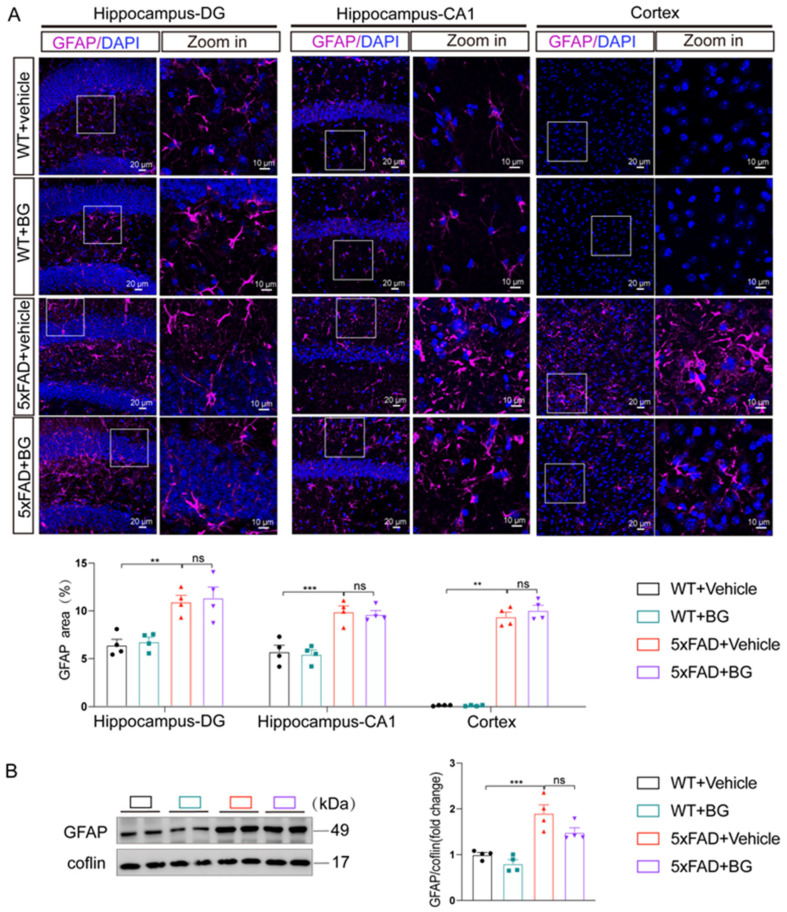
BG did not reduce activated astrocytes in 5×FAD mice. (**A**) Representative images of GFAP-labeled astrocytes and quantification of immunoreactivity area of GFAP in hippocampal DG region, CA1 region, and cortex of WT + vehicle, WT + BG, 5×FAD + vehicle and 5×FAD + BG, respectively (*n* = 4 mice per group).The white frame represents the area to be zoomed. (**B**) Representative immunoblots and quantitative analyses of GFAP (*n* = 4 per group). Data are mean ± SEM. ** *p* < 0.01 and *** *p* < 0.001; ns, not significant. One-way ANOVA, followed by Tukey’s multiple comparisons test (left and middle panel in **A**,**B**). Welch’s ANOVA test, followed by Dunnett’s T3 multiple comparisons test (right panel in **A**).

**Figure 6 nutrients-17-03218-f006:**
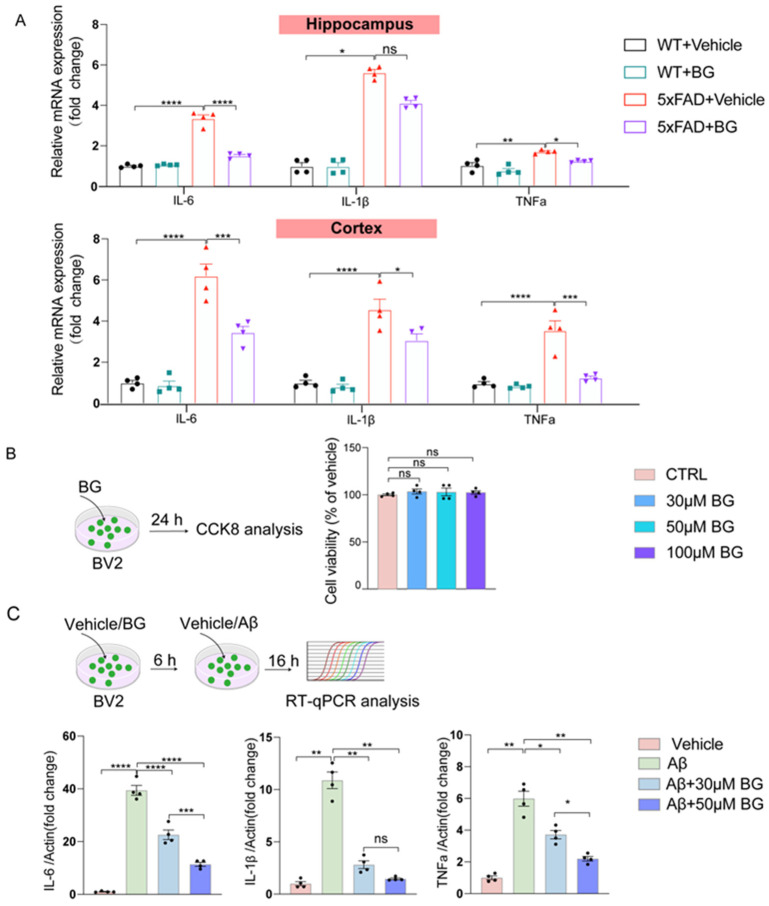
BG reduced pro-inflammatory cytokine levels in 5×FAD mice and Aβ-stimulated BV2 cells. (**A**) The mRNA expression levels of Tnfα, Il-1β, and Il-6 in WT + vehicle, WT + BG, 5×FAD + vehicle and AD + BG, respectively (*n* = 4 mice per group). (**B**) BV2 cells were treated with or without different concentrations of BG for 24 h, and the cell viability was measured by CCK-8 assay (*n* = 4 per group). (**C**) Left, experimental schematic of relative pro-inflammatory cytokine expression levels by RT-qPCR analysis after different treatment in BV2 cultures. Right, the mRNA expression levels of Tnfα, Il-1β, and Il-6 in BV2 cell under differing treatment conditions (*n* = 4 per group). Data are mean ± SEM. * *p* < 0.05, ** *p* < 0.01,*** *p* < 0.001 and **** *p* < 0.0001; ns, not significant. One-way ANOVA, followed by Tukey’s multiple comparisons test ((**A**) except the analysis of Il-1β expression in hippocampus, left panel in (**C**)). Welch’s ANOVA test, followed by Dunnett’s T3 multiple comparisons test ((**B**), middle and right panel in (**C**)). Kruskal–Wallis test, followed by Dunn’s multiple comparisons test (the analysis of Il-1β expression in hippocampus in (**A**)).

**Figure 7 nutrients-17-03218-f007:**
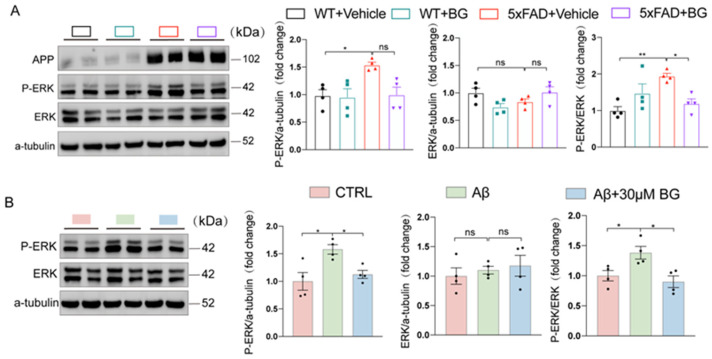
BG regulated MAPK signaling pathways in AD or Aβ-stimulated BV2 cells. (**A**) Representative immunoblots and quantitative analyses of P-ERK and ERK in the hippocampus of WT + vehicle, WT + BG, 5×FAD+ vehicle and 5×FAD + BG (*n* = 4 mice per group). (**B**) Representative immunoblots and quantitative analyses of P-ERK and ERK in BV2 cell treated with vehicle, Aβ+vehicle and Aβ+30 μM BG (*n* = 4 per group). Data are mean ± SEM. * *p* < 0.05 and ** *p* < 0.01; ns, not significant. One-way ANOVA, followed by Tukey’s multiple comparisons test (middle and right panel in (**A**,**B**)). Welch’s ANOVA test, followed by Dunnett’s T3 multiple comparisons test (left panel in (**B**)).

## Data Availability

The raw data supporting the conclusions of this article will be made available by the authors on request.

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
