# Peer review of "Natural Small-Molecule Bergapten Ameliorates Amyloid-β Pathology and Neuroinflammation in Alzheimer’s Disease"

_nutrients, 2025, doi:10.3390/nu17203218_

Round 1

Reviewer 1 Report

Comments and Suggestions for Authors

In this study, the beneficial effects of bergapten on behavior and neuropathology of 5xFAD mice are investigated, supported by bioinformatic analyses pointing to the MAPK pathway as a potential target of bergapten, and further validated through molecular analyses. The study is carefully designed, the data are clearly presented, and the manuscript is well written.

I only have a few minor comments:

Line 291: The term “validated” does not seem appropriate in this context. Alternatives such as “suggest” or “indicate” would be more accurate, unless e.g. functional assays are provided to directly confirm the effect.

Figure 3: In panel E, “5xFAD” is used, whereas other panels and figures use “AD”. Please ensure consistency across figures. Since the text predominantly refers to 5xFAD, this may be the preferable choice.

Limitations section: It would strengthen the manuscript to discuss  what is already known about bergapten. For instance, phototoxicity is a well-documented side effect, and several other pathways modulated by bergapten have been described in the literature. Including these points would better contextualize your findings.

Future directions: The authors could also suggest experiments that would clarify whether the observed effects of bergapten on MAPK signaling are causal for the beneficial outcomes in AD pathology.

Reviewer 2 Report

Comments and Suggestions for Authors

This is a promising study on bergapten (BG) as a potential treatment for Alzheimer's disease (AD).

The study only administered BG for 30 consecutive days to six-month-old 5×FAD mice. While this period showed behavioral improvements and reduced pathology, AD is a chronic, progressive neurodegenerative disease. A longer-term study is crucial to determine if BG can sustain these benefits or if the effects wane over time. A 30-day intervention may not be representative of a preventative or long-term therapeutic strategy.

The study highlights the MAPK signaling pathway as the "top-ranked" one from bioinformatics analysis. While this is a good starting point, bioinformatics can sometimes generate false positives or rank pathways based on statistical significance without considering biological relevance. 

he study was conducted on 5×FAD mice, a widely used but imperfect model of AD. While these mice show rapid Aβ accumulation and cognitive deficits, their pathology doesn't fully mimic the complex, multi-factorial nature of human AD. The results from a mouse model, especially with such a short intervention, do not guarantee similar effects in humans.

The introduction mentions that dietary factors play a significant role in AD, and BG is a natural compound. However, the study administered BG via intragastric administration (directly into the stomach via a tube), which bypasses the natural complexities of digestion and absorption from food. This method doesn't reflect how BG would be consumed as part of a regular diet, making the results less directly applicable to a dietary intervention strategy.

The study's in vitro experiments used Aβ-stimulated BV2 microglial cells. While these cells are a standard tool for studying neuroinflammation, they are an immortalized cell line, not primary microglia. Their behavior may not perfectly mirror the complex, in vivo interactions of microglia in the brain, especially in the context of other cell types like astrocytes and neurons.
